# Evaluation of Shock Tube Retrofitted with Fast-Opening Valve for Dynamic Pressure Calibration

**DOI:** 10.3390/s21134470

**Published:** 2021-06-29

**Authors:** Eynas Amer, Mikolaj Wozniak, Gustav Jönsson, Fredrik Arrhén

**Affiliations:** Measurement Science and Technology, RISE Research Institutes of Sweden, 504 62 Borås, Sweden; mikolaj.wozniak@ri.se (M.W.); gustav.jonsson@ri.se (G.J.); fredrik.arrhen@ri.se (F.A.)

**Keywords:** dynamic pressure, shock tube, fast-opening valve, repeatability

## Abstract

Accurate dynamic pressure measurements are increasingly important. While traceability is lacking, several National Metrology Institutes (NMIs) and calibration laboratories are currently establishing calibration capacities. Shock tubes generating pressure steps with rise times below 1 μs are highly suitable as standards for dynamic pressures in gas. In this work, we present the results from applying a fast-opening valve (FOV) to a shock tube designed for dynamic pressure measurements. We compare the performance of the shock tube when operated with conventional single and double diaphragms and when operated using an FOV. Different aspects are addressed: shock-wave formation, repeatability in amplitude of the realized pressure steps, the assessment of the required driver pressure for realizing nominal pressure steps, and economy. The results show that using the FOV has many advantages compared to the diaphragm: better repeatability, eight times faster to operate, and enables automation of the test sequences.

## 1. Introduction

Dynamic pressure measurements are increasingly important to achieve accurate process control. Examples include applications such as medical equipment, turbomachines [1], and combustion engines [2], where the accuracy in pressure measurements influences safety, emissions, and economy.

While traceable dynamic calibrations are lacking [3], several National Metrology Institutes (NMIs) and calibration laboratories are working to establish a traceability chain [4]. At the national laboratory for pressure in Sweden, a future standard for dynamic pressures based on shock tube technology is under development. Shock tubes can realize pressure steps with rise time well below 1 μs and are preferred when realizing high-frequency dynamic reference pressures [5,6,7]. In calibration, shock tubes may either be used as a pure pressure generator in combination with a reference sensor or be used as a primary measurement standard [8]. Both applications, however, require the shock tube to produce repeatable high-quality shocks to be relevant.

Traditionally, shocks have been initiated by the bursting of diaphragms [9]. This method often results in high quality shocks but is tedious for the operator. Diaphragms need to be exchanged between each shock realization. Fragments of ruptured diaphragms involve a risk of damaging the sensors, including the test object, and must be removed from the shock tube after each test. In effect, the use of diaphragms requires manual work to produce multiple shocks, and the repeatability is limited.

Fast-opening valves pose an alternative approach to initiate shocks [10]. Recently they have been applied to metrological purposes [11,12]. In principle, the fast-opening valves (FOVs) require no manual work to operate and do not pose any danger to the test objects due to debris. On the downside, FOVs typically have a considerably longer opening time than bursting diaphragms [10], thus putting new constraints on the shock-tube design.

In this work, we evaluate an installation of a fast-opening valve with focus on precision and repeatability of the generated pressure steps and compare the results with two alternative methods to initiate shocks, using diaphragms. We also note the time required to make two consecutive shocks, using the different methods. The operating time directly affects the economy of the system and, thus, the viability of this future dynamic pressure standard. A fast-opening valve furthermore enables fully automated operation of the shock. This is, however, outside the scope of this work.

## 2. Experimental Setup

The work was carried out using the shock tube schematically depicted in Figure 1. The shock tube consists of two parts: the driven section, filled with low-pressure driven gas, separated by a fixture for diaphragm or by a FOV (KB-80-20, ISTA Pneumatics, Saint-Petersburg, Russia) from the driver section that contains a high-pressure driver gas. The two sections have an inner diameter of 100 mm and lengths L_1_ and L_2_, respectively. Three different operating modes; single and double diaphragm and FOV were used in this work. In case of diaphragm operating modes, the nominal length of the driven section L_1_ was 3 m, and the driver section had a length L_2_ of 2 m. In the case of using the FOV as an operating mode, L_1_ was 7 m, and L_2_ was 3 m.

The driven section is equipped with piezo electrical (PE) pressure sensors (113A21, PCB, New York, NY, USA) flush mounted at well-defined positions on the circumference along a straight line parallel to the central axis of the shock tube. These sidewall sensors allow for both monitoring of shock formation and measuring of shock propagation speed. An identical PE pressure sensor was also flush mounted in the center of the endwall. The PE sensors were connected to a signal conditioner (Model 483C Series, PCB, New York, NY, USA) and digitized using an 8 channel 12 bit 60 MS/s per channel oscilloscope (PXI-5105, NI, Austin, TX, USA). In this work, the data acquisition was performed by using LabVIEW with a sampling rate of 3 MS/s and a sample size of 10^5^ samples. The sensitivity of individual sensors given by the manufacturer were used to scale the recorded pressure amplitude.

Static absolute pressure transmitters were mounted on the driven section (EJX 510A, Yokogawa, Tokyo, Japan) and the driver section (EJX 310A, Yokogawa, Tokyo, Japan) to monitor the initial pressures of these volumes. The buffer pressure, between the double diaphragms, was monitored by using a static gauge pressure transmitter (EJX 530A, Yokogawa, Tokyo, Japan). As the shock tube is positioned in a temperature-controlled laboratory, the initial gas temperature was considered to be 21 °C.

The driver section, driven section, and buffer volume (in the case of double diaphragms) can be filled, vented, and evacuated independently. To prevent over-filling, the buffer volume is filled by gas from the driver section. Two separate gas mixtures may be routed to the different volumes. In this work, Ar (99.999%, Air Liquide) was used in all volumes for pressure steps up to 500 kPa. For pressure steps above 500 kPa, Ar was used in the driven section and He (99.999%, Air Liquide) in the driver. To ensure well-known gas composition in driver and driven sections, the system is equipped with a dry roots vacuum pump (NeoDry 15E, Kashiyama, Saku City, Japan) to evacuate the volumes before filling with appropriate gases.

When using the single diaphragm mode, the driver volume was pressurized until the single diaphragm spontaneously ruptured. For double diaphragms, the shock is initiated by venting the buffer volume. The diaphragms used were either a single ply, or a combination, of 23, 36, or 50 μm Mylar foil.

The resulting shock amplitudes were calculated from the measured shock propagation speed and static initial conditions, using a 1D transport model described in Section 3.

## 3. Theory

The amplitude of a 1D shock-wave propagating in a calorically perfect gas (having constant specific heats with respect to temperature) can be expressed as follows [13]:(1)$$\frac{p_{2}}{p_{1}}=1+\frac{2\gamma_{1}}{\gamma_{1}+1}\left(M_{s}^{2}-1\right)$$
where *p_2_* denotes the pressure behind the shock front, *p*_1_ the pressure in front of the shock, *γ*_1_ is the heat capacity ratio *c_p_*/*c_v_* of the driven gas, and *M*_s_ is the Mach number of the incident shock given by the following:(2)$$M_{s}=\frac{W}{a_{1}}$$
where *W* denotes the speed of the shock front relative to the net speed of the gas molecules in front of the shock, and $a_{1}=\sqrt{\gamma_{1}R_{1}T_{1}}$ is the speed of sound in the undisturbed driven gas at absolute temperature, *T*_1_. *R*_1_ is the specific gas constant of the driven gas.

Equations (1) and (2) are straightforwardly applied to the incident shock wave (i.e., directly after the rupture of diaphragms or opening of the FOV and when propagating in a driven gas at rest). The quantitative values of *p*_1_ and *T*_1_ are given by static measurements of the driven gas. *W* is calculated from the delay between the measured arrival of the shock front at the sidewall mounted PE sensors and knowledge of their positions.

Equation (1) can also be applied to the reflected shock amplitude provided supplementary relations. The reflected shock propagates in the opposite direction through the tail of the incident shock. Hence, for reflected shocks, *p*_2_ and *p*_1_ in Equation (1) instead denote the reflected amplitude (*p*_5_) and the pressure behind the incoming shock front (*p*_2_), respectively. The Mach number of the reflected shock (*M*_R_) is connected to the Mach number of the incident shock (*M*_s_) by the implicit expression [13]:(3)$$\frac{M_{R}}{M_{R}^{2}-1}=\frac{M_{s}}{M_{s}^{2}-1}\sqrt{1+\frac{2\left(\gamma_{1}-1\right)}{{\left(\gamma_{1}+1\right)}^{2}}}\left(M_{s}^{2}-1\right)\left(\gamma_{1}+\frac{1}{M_{s}^{2}}\right)$$

Hence, the amplitude of the reflected shock wave can be obtained as follows:(4)$$\frac{p_{5}}{p_{2}}=1+\frac{2\gamma_{1}}{\gamma_{1}+1}\left(M_{R}^{2}-1\right)$$

From Equation (4) and the measured driven gas static pressure, *p*_1_, the pressure step amplitude (*p*_5_–*p*_1_) can be calculated. When employing a shock tube as a dynamic pressure standard, the device under test is preferably flush-mounted on the endwall to record the reflected pressure step (*p*_5_–*p*_1_), as this results in the highest amplitude and fastest rise-time of the pressure step.

The eventual Mach number of the incident shock partly depends on the conditions of the undisturbed driver gas (composition, temperature, and pressure). The relation between the pressure ratio of the undisturbed driver and driven gases, *p*_4_/*p*_1_, and the Mach number, *M*_s_, is given by the following [14]:(5)$$\frac{p_{4}}{p_{1}}=\frac{1}{\alpha_{1}}\left(\frac{2\gamma_{1}M_{s}^{2}}{\gamma_{1}-1}-1\right){\left(1-\frac{\left(1/\alpha_{4}\right)\left(a_{1}/a_{4}\right)\left(M_{s}^{2}-1\right)}{M_{s}}\right)}^{-2\gamma_{4}/\left(\gamma_{4}-1\right)}$$
where *α*_1_ = (*γ*_1_ + 1) / (*γ*_1_−1), *α*_4_ = (*γ*_4_ + 1) / (*γ*_4_−1), *γ_4_* is the heat capacity ratio, *c_p_*/*c_v_*, of the driver gas, and *a*_4_ is the speed of sound in the undisturbed driver gas.

For realizing a nominal Mach number (*M*_s_), Equation (5) can be used to estimate a preliminary value of the required driver pressure (*p*_4_) at a given driven pressure (*p*_1_). Deviation between experimental and estimated values of *p*_4_ and possible reasons for it are discussed in the next section.

## 4. Results and Discussion

A comparison of the three operating modes with respect to shock formation, estimation of the required driver pressure (*p*_4_), repeatability, and economy is presented and discussed.

### 4.1. Shock Formation

Typical shock waves generated by the shock tube are shown in Figure 2. Figure 2a presents the shock wave generated by bursting the double diaphragm mode. Shock waves generated by the FOV at two different lengths of the shock tube, 5 and 10 m, are shown in Figure 2b,c, respectively. For all three realizations, the nominal pressure step was 500 kPa and the driven pressure was kept at 100 kPa(a). The shock waves were recorded by the endwall sensor and the sidewall sensors 1–3. Figure 2a shows that the measured amplitude of the generated pressure step is about 500 kPa and that the pressure remains constant during a period of about 4 ms before it interacts with the contact surface between the driver and driven gases. Figure 2 shows a similar behavior when using the FOV albeit that the generated shock front is not fully developed, resulting in a non-constant pressure during the high-pressure part of the pressure step. The length of the driven section (L_1_ = 3 m) is not enough for the formation of the shock wave at the present opening speed of the valve. This can be seen in Figure 3a, comparing the shock profiles at three locations along the shock tube. The pressure traces in the figure are recorded by sidewall sensors 6, 4, and 1, which are located at distances of about 1.4, 2.2, and 2.8 m, respectively, from the FOV (c.f. Figure 1). The pressure profiles were shifted in time for ease of comparison. The shock front is progressively converging from a softer transition close to the FOV into a more ideal step further down the shock tube. We argue that the opening time of the FOV is considerably longer than the rupture time of diaphragms that will lead to a slower development of the shock front. As illustrated in Figure 2b and Figure 3a, this relatively slow opening time results in not fully developed shocks at the end of the 3 m driven section. Furthermore, the shock speed along the tube was calculated at different positions, using the sidewall sensors. The distance between the sidewall sensors and the FOV is *x_i_*, and the arrival times of the shock front at the sensors positions is *t_i_*, *i* = [1 − 6]. The shock speed at the middle distance between every two successive sensors is *W_i_* = (*x_i + 1_* − *x_i_*) / (*t_i + 1_* − *t_i_*). Figure 3b shows the shock speed at different positions from the FOV. The figure shows that the shock wave is accelerating when propagating along the shock tube that confirms that the shock was not fully developed when reached the back plate. To generate a fully developed shock, the driven section was elongated to 7 m. The generated shock wave after elongating the shock tube is shown in Figure 2c. A shock-wave profile similar to that generated using diaphragms was obtained. The shock speed measured by the sidewall sensors at different positions after elongating the shock tube is presented in Figure 4. The deceleration of the shock wave as it propagates toward the back plate—this is the expected behavior of fully developed shocks [13]—can be seen in the figure.

Furthermore, the driver section was elongated to 3 m. When using He as a driver gas, the time duration of the constant pressure step is shortened as the expansion wave moves faster than the shock wave. A short constant pressure limits the minimum sensor response frequency that can be characterized by the shock tube. To generate reasonable constant pressure steps, the driver section was elongated, and this delayed the arrival of the expansion wave.

These settings (L_1_ = 7 m and L_2_ = 3 m) were used in all FOV experiments.

The effects of different operating modes on the pressure step parameters, namely amplitude, step duration, and rise time, were investigated. There is no difference between the three operating modes for realizing specific step amplitude (*p*_5_–*p*_1_) at specific Mach number, provided that the proper initial conditions are used (see Figure 2a,c). The step duration depends on the length of the shock tube, the used gases in different sections (driven and driver), and the operating mode. Qualitatively, the same design rules are applicable for different operating modes. However, we did not investigate it quantitively. The rise time of the generated step is not possible to be measured in the current setup. However, we have no reason to believe that it is affected by the operating mode, as long as the generated shock is fully developed.

### 4.2. Driver Pressure (p_4_) and Mach Number

When realizing the pressure steps, we observed that the required driver pressure differs from the ideal value given by Equation (5). The behavior of this discrepancy differs between various operating modes. When using the diaphragm modes (single and double), the value of (*p*_4_ experiment/*p*_4_ equation) was constant for all pressure steps and driver gas compositions. It is 1.1 in case of single diaphragm and 1.2 for double diaphragm. In contrast, in the case of using the FOV, it changes with the Mach number and the composition of the driver gas. Figure 5a and Figure 6a show the behavior of *p*_4_ from the experiment using FOV and calculated from Equation (5) at different Mach numbers when using Ar and He as a driver gas, respectively. The value of (*p*_4_ experiment/*p*_4_ equation) was calculated at different Mach numbers when using Ar and He as a driver gas and is shown in Figure 5 and Figure 6b, respectively. In all cases, *p*_1_ was kept at 100 kPa(a). The Mach number was calculated by using Equation (2), where the shock speed was calculated by using the linear regression of the time–position relation measured by the sidewall sensors 1–3 shown in Figure 1.

The figures show that ratio (*p*_4_ experiment/*p*_4_ equation) increases linearly with the Mach number for both Ar and He as a driver gas. The slope of the line depends on the driver gas properties (heat capacity ratio and molecular weight). When using Ar as a driver gas, $p_{4\;\mathrm{ratio}}=0.98\times M_{s}+0.19$, while in case of using He, $p_{4\;\mathrm{ratio}}=0.57\times M_{s}+0.40$. We argue that the geometry of the interface between driver and driven sections affects the relation between driver pressure and Mach number. While the geometry for the diaphragm fixtures is constant with respect to both pressure and time after burst, the geometry of the valve is varying with both pressure and time. The valve is spring-loaded in its closing direction, and the opening is partially actuated by the driver pressure (*p*_4_). Therefore, both the speed and stroke of the valve stem and disc upon valve opening are depending on *p*_4_. This naturally results in a more complicated relationship between *p*_4_ and *M*_s_.

Since the FOV required higher driver pressure compared to the diaphragm, the accessible *M*_s_ and the resulting pressure steps may be limited by the highest allowable driver pressure. The accuracy of the assessed pressure steps is not affected by the driver pressure if *M*_s_ or a reference sensor is used. Thus, we strongly recommend not to use *p*_4_ to assess the pressure steps.

### 4.3. Repeatability

Figure 7 presents the repeatability of the generated pressure steps, using the three different operating modes. Three repetitions of five nominal pressure steps (*p*_5_–*p*_1_) ranging between 200 and 1000 kPa were realized, using each mode of operation. The difference between the maximum and minimum values at each pressure level and operating mode were calculated and are shown in the figure.

Figure 7 shows that the repeatability realized when using the FOV is better than that in the case of using the diaphragm modes. In absolute numbers, the repeatability is within ±4 kPa for FOV compared with ±16 kPa for diaphragm modes. At low pressure steps up to 300 kPa, the relative repeatability of the FOV is within ±1% and improved to ±0.6% for pressure steps above 300 kPa. On the other hand, in the case of using diaphragm modes, the relative repeatability is within ±3.7% up to 300 kPa and within ±2.2% for pressure steps above 300 kPa.

Sources for variations in shock amplitude common for all modes of operation comprise variations in initial parameters, such as driven pressure (*p*_1_), driver pressure (*p*_4_), temperature of driven gas (*T*_1_), and driver gas (*T*_4_) and purity of respective gases. These variations put a lower limit on the repeatability common for all modes, and their influence on the final pressure step can be estimated by using Equations (1)–(5). Theses equations imply that ± 1 °C variations in *T*_1_ and *T*_4_ individually affect the pressure step by ± 1 kPa. An observed ± 0.1% variation in *p*_4_/*p*_1_ affects the step amplitude by ± 0.2 kPa.

Additional effects from diaphragm rupture or valve opening appear to be present as the observed variations in pressure step exceed the projection from variations in the input parameters.

For single diaphragm operation, the driver pressure is determined by the ultimate pressure the individual diaphragm can withstand before rupture. That pressure is in turn determined by the material properties of the individual diaphragms, as well as any irregularities and strains induced by the operator upon mounting the diaphragms. The dependence of the bursting pressure, *p*_4_, on the diaphragm thickness is presented in detail in Figure 8. Here different diaphragm thicknesses ranging from 23 to 200 μm were tested. The driven pressure (*p*_1_) was kept at 100 kPa(a) in all measurements. The driver pressure was increased gradually until the diaphragm burst. Three measurements were taken at every thickness. The figure shows that the driver pressure increases linearly from 200 kPa(a) at a diaphragm thickness of 23 μm to about 1000 kPa(a) at 200 μm. In this study, only three different ply thicknesses were used to build a diaphragm, resulting in that only a selection of discrete levels in *p*_4_ is achievable. As can be seen in Figure 8, there is only a practical problem for lower pressures as the variance in *p*_4_ becomes dominant at higher pressures.

When double-diaphragm mode was applied, the problems with variations in driver pressure were, to a large extent, mitigated, as the ruptures of the diaphragms are initiated by venting of the buffer volume. This method allows for better control of the initial parameters. The double-diaphragm mode is expected to have considerably better repeatability than the single diaphragm; this also proves to be the case according to Figure 7. However, the irregularities and mounting effects are still unavoidable.

The FOV allows for better control over the initial pressures. We argue that the repeatability of the shocks generated by the FOV is mainly limited by uncertainties of the variations of the gas temperature and the opening behavior of the valve.

### 4.4. Economy

We used this measure to estimate the time needed to perform standard calibrations. A comparison of the operation time of the three modes is shown in Figure 9. Even without automation, the FOV mode requires one-fifth of the operation time of the single diaphragm and one-eighth of the double-diaphragm operation time.

The additional time for operating the diaphragms stems from disassembly and reassembly of the fixture for the diaphragms and cleaning the shock tube from debris. In the double-diaphragm mode, the changing of diaphragms is time-consuming, and there is also more debris to clean compared to the single-diaphragm mode. The setting of initial pressures is also more time-consuming for double diaphragms, as care must be taken to maintain a correct pressure gradient over the double diaphragms in order to avoid accidental rupture.

Furthermore, using the FOV eliminates the risk of damaging the sensors, including the test object, due to debris from the ruptured diaphragm in the driven section.

## 5. Summary

In this work, the performance of a shock tube retrofitted with a fast-opening valve (FOV) was evaluated. The shock tube is the candidate of the Swedish national laboratory for pressure as a primary measurement standard for dynamic pressure calibration. Shock waves were initiated by spontaneous rupture of single diaphragm, controlled rupture of double diaphragm, and an FOV. A comparison of the three operating modes was performed. The FOV allows for the realization of shocks with a better repeatability that is comparable to diaphragm operation modes and with a repetition rate up to eight times faster.

## Figures and Tables

**Figure 1 sensors-21-04470-f001:**
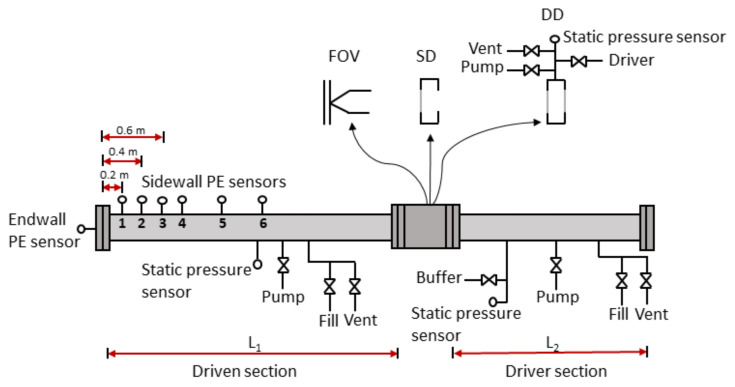
Schematic illustration of the shock tube. FOV, fast-opening valve; SD, single diaphragm; DD, double diaphragm.

**Figure 2 sensors-21-04470-f002:**
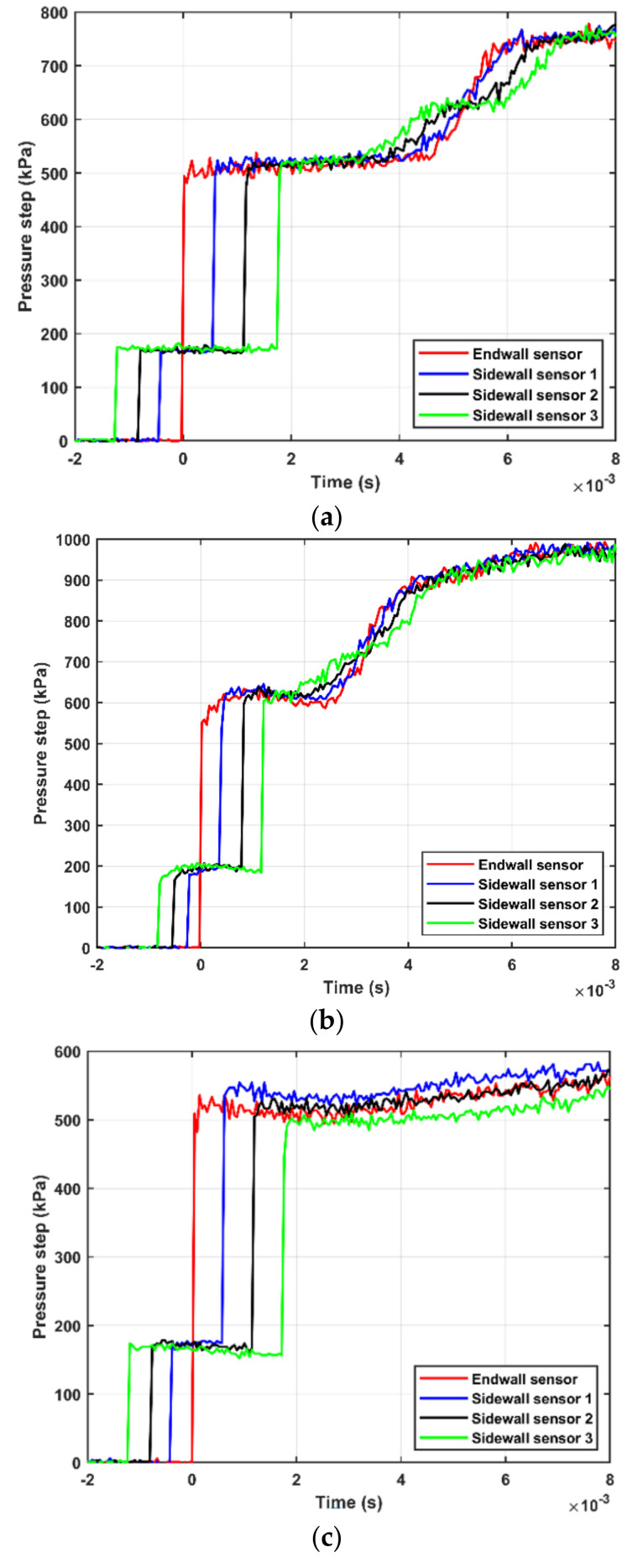
A typical shock wave generated by the shock tube. (**a**) Double diaphragms; (**b**,**c**) fast-opening valve before and after elongating the shock tube, respectively.

**Figure 3 sensors-21-04470-f003:**
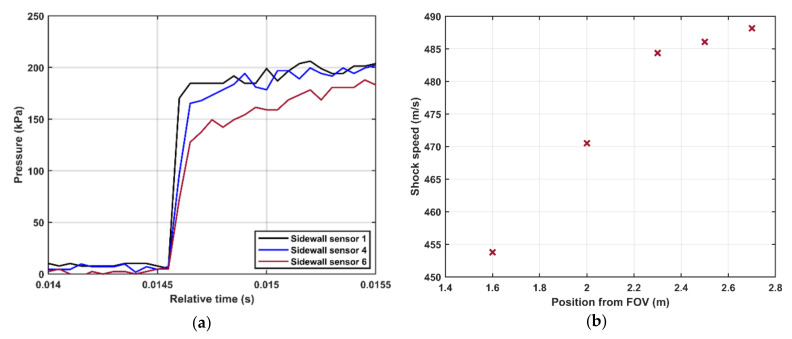
(**a**) The development of the incident shock front generated by the FOV at three different positions. (**b**) The shock speed at different positions from the FOV. Driven section length (L_1_) is 3 m.

**Figure 4 sensors-21-04470-f004:**
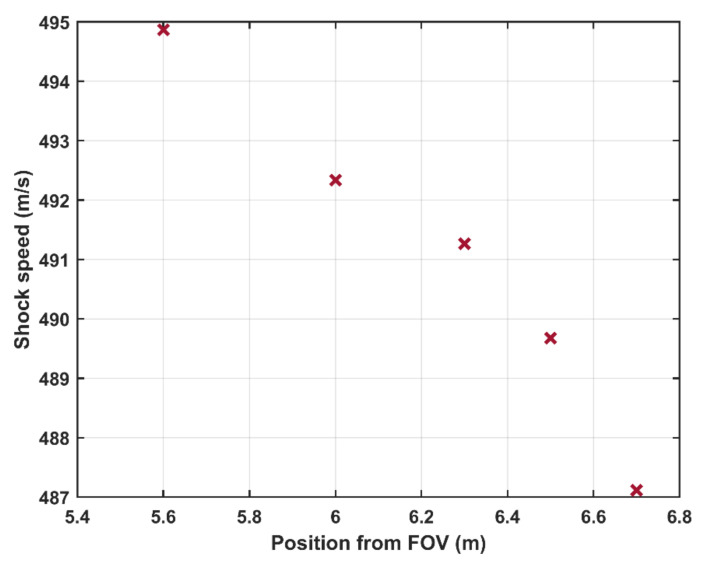
The shock speed at different positions from the FOV. Driven section length (L_1_) is 7 m.

**Figure 5 sensors-21-04470-f005:**
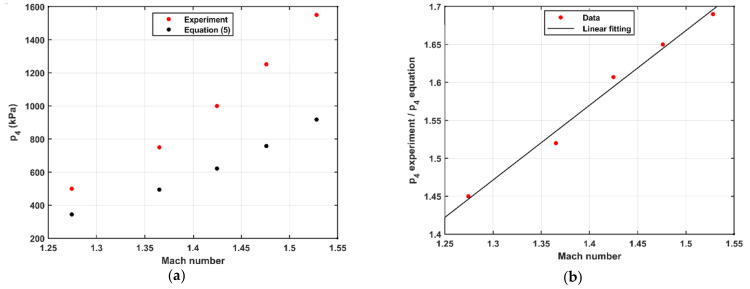
(**a**) The driver pressure (*p*_4_) as a function of Mach number and (**b**) (*p*_4_ experiment/*p*_4_ equation) as a function of Mach number when using Ar as a driver gas; *p*_1_ is 100 kPa(a).

**Figure 6 sensors-21-04470-f006:**
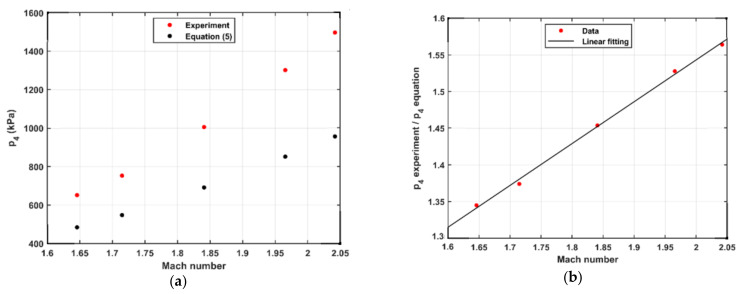
(**a**) The driver pressure (*p*_4_) as a function of Mach number and (**b**) (*p*_4_ experiment/*p*_4_ equation) as a function of Mach number when using He as a driver gas; *p*_1_ is 100 kPa(a).

**Figure 7 sensors-21-04470-f007:**
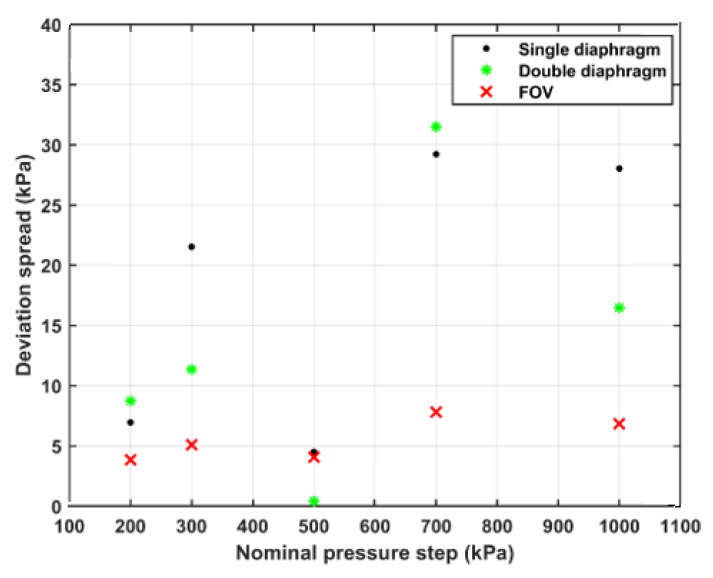
The deviation spread at different nominal pressure steps, using the three operating modes.

**Figure 8 sensors-21-04470-f008:**
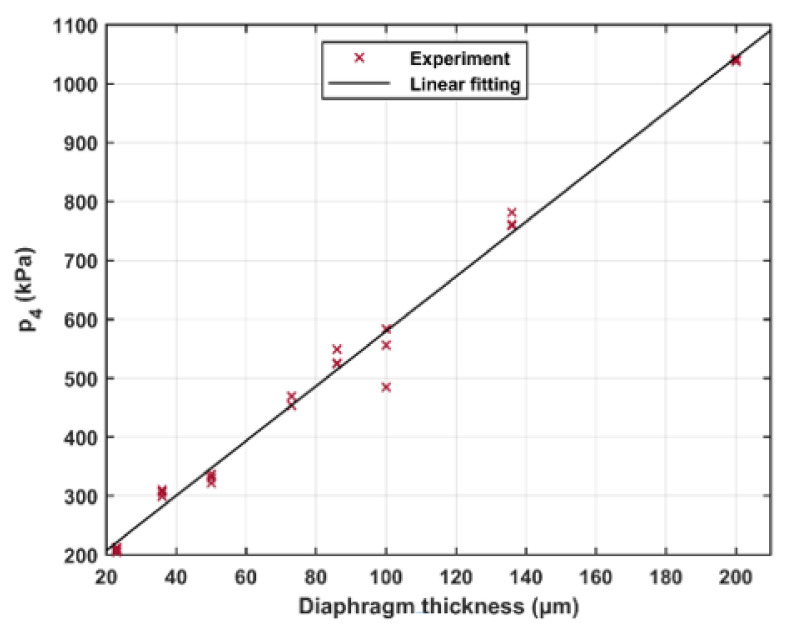
The bursting pressure of the single diaphragm as a function of the diaphragm thickness; *p*_1_ is 100 kPa(a).

**Figure 9 sensors-21-04470-f009:**
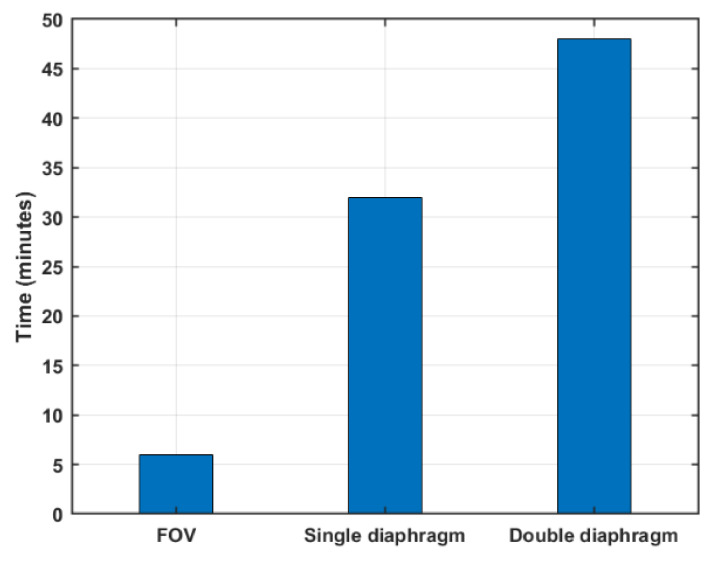
A comparison of the operation time of the three operating modes.

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
