# Peer review of "Evaluation of Shock Tube Retrofitted with Fast-Opening Valve for Dynamic Pressure Calibration"

_sensors, 2021, doi:10.3390/s21134470_

Round 1
Reviewer 1 Report
The authors present an experimental shock-tube set up, with fast-opening valves and show the practical feasibility in terms of experiment time, repeatability as well as ease of set-up with respect to standard diaphragm based set-ups. The paper is well written and the results are well-presented, objectively. The only comment I have is regarding the deviation of the post-shocked pressure from the Rankine-Hugoniot jump conditions. The authors in Fig 5 and 6 show (and present reasonable conjectures) that there is a systematic bias between the RH derived pressures and the measured pressures. It may be worthwhile to quantify this linear equation explicitly, as at the very least it gives the users a tool for knowing the uncertainty in using the FOV based approach. Additionally, it may also be reasonable to provide some reasonable arguments/conjectures towards minimizing the mismatches between the RH based relations and the experimentally observed pressures.
Reviewer 2 Report
This manuscript presented the results from applying a fast-opening valve to a shock tube designed for dynamic pressure measurements. This work is of important in metrology and measurement of dynamic pressure in many applications. Some issues need to be addressed before acceptation.
- In section 4.2, how to calculate the incident shock Mach number? Using two sensors? Or more?
- The reflected step pressure parameters affecting the calibration range of shock tube are step amplitude, rise time and time of duration. This paper should investigate the effect of diaphragm type (single- and double diaphragm, and fast-opening valve) on these parameters of reflected step pressure generated in shock tube.
- More references relate to shock tube should be cited and some figures require improvement, such as figure 2.
